**Subject Category:**
Biology (whole organism)

evolution/cognition

evolution, multilevel selection, transitions in individuality, Bayesian models, structure learning

**Author for correspondence:**
Dániel Czégel
e-mail: danielczegel@gmail.com

# Multilevel selection as Bayesian inference, major transitions in individuality as structure learning

Dániel Czégel[1,2,4,5], István Zachar[1,3,4]
and Eörs Szathmáry[1,2,4]

[1]MTA Centre for Ecological Research, Evolutionary Systems Research Group, Hungarian Academy of Sciences, 8237 Tihany, Hungary
[2]Department of Plant Systematics, Ecology and Theoretical Biology, and [3]MTA-ELTE Theoretical Biology and Evolutionary Ecology Research Group, Eötvös University, 1117 Budapest, Hungary
[4]Parmenides Foundation, Center for the Conceptual Foundations of Science, 82049 Pullach/Munich, Germany
[5]Department of Brain and Cognitive Sciences, Massachusetts Institute of Technology, Cambridge, MA 02139, USA

DC, 0000-0002-5722-1598; IZ, 0000-0002-3505-0628; ES, 0000-0001-5227-2997

Complexity of life forms on the Earth has increased tremendously, primarily driven by subsequent evolutionary transitions in individuality, a mechanism in which units formerly being capable of independent replication combine to form higher-level evolutionary units. Although this process has been likened to the recursive combination of pre-adapted sub-solutions in the framework of learning theory, no general mathematical formalization of this analogy has been provided yet. Here we show, building on former results connecting replicator dynamics and Bayesian update, that (i) evolution of a hierarchical population under multilevel selection is equivalent to Bayesian inference in hierarchical Bayesian models and (ii) evolutionary transitions in individuality, driven by synergistic fitness interactions, is equivalent to learning the structure of hierarchical models via Bayesian model comparison. These correspondences support a learning theory-oriented narrative of evolutionary complexification: the complexity and depth of the hierarchical structure of individuality mirror the amount and complexity of data that have been integrated about the environment through the course of evolutionary history.

## 1. Introduction

On Earth, life has undergone immense complexification [1,2]. The evolutionary path from the first self-replicating molecules to

structured societies of multicellular organisms has been paved with exceptional milestones: units that were capable of independent replication have combined to form a higher-level unit of replication [3–5]. Such *evolutionary transitions in individuality* opened the door to the vast increase of complexity. Paradigmatic examples include the transition of replicating molecules to protocells, the endosymbiosis of mitochondria and plastids by eukaryotic cells and the appearance of multicellular organisms and eusociality. Interestingly, it is possible to identify common evolutionary mechanisms that possibly led to these unique but analogous events [6–9]. A crucial preliminary condition is the *alignment of interests*: to undergo an evolutionary transition in individuality, organisms must exhibit cooperation, originating from genetic relatedness and/or synergistic fitness interactions [4]. However, the story does not end here: something must also maintain the alignment of interests subsequent to the transition. At any phase, the fate of the organism depends on selective forces at multiple levels that might be in conflict with each other. Incorporating the effects of multilevel selection is, therefore, a crucial element of understanding evolutionary transitions in individuality [10].

These theoretical considerations above delineate conditions under which a transition might occur and a possibly different set of conditions which help to maintain the integrity of units that have already undergone transition. However, these considerations alone cannot offer a predictive theory of complexification as they do not address the question of how necessary these environmental and ecological conditions are. An alternative, supplementary approach that circumvents these difficulties is to investigate whether mathematical theories of adaptation and learning can provide further insights about the general scheme of evolutionary transitions in individuality. In this paper, we argue that they do. We first provide a mapping between multilevel selection modelled by discrete-time replicator dynamics and Bayesian inference in belief networks (i.e. directed graphical models), which shows that the underlying mathematical structures are isomorphic. The two key ingredients are (i) the already known equivalence between univariate Bayesian update and single-level replicator dynamics [11,12] and (ii) a possible correspondence between properties of a hierarchical population composition and multivariate probability theory. We then show that this isomorphism allows for a natural interpretation of evolutionary transitions in individuality as *learning the structure* [13,14] of the belief network. Indeed, following adaptive paths on the fitness landscape over possible hierarchical population compositions is equivalent to a well-known method used for selecting the optimal model structure in the Bayesian paradigm, namely, *Bayesian model comparison*. This suggests that complexification of life via successive evolutionary transitions in individuality is analogous to the complexification of optimal model structure as more (or more complex) data about the environment is available. These ideas are illustrated in figure 1; for more details, see Methods and Results.

Relating the dynamics of evolutionary complexification to hierarchical probabilistic generative models complements recent efforts of searching for algorithmic analogies between emergent evolutionary phenomena and neural network-based learning models [15,16]. These include correspondences between evolutionary-ecological dynamics and autoassociative networks [17] and also linking the evolution of developmental organization to learning in artificial neural networks [18]. As such connectionist models account for how global self-organizing learning behaviour might emerge from simple local rules (e.g. weight updates), our approach aims at providing a common global framework for modelling both evolutionary and learning dynamics.

## 1.1. Darwinian evolution of multilevel populations

Populations of replicators, like genes, chromosomes and cells, assemble into hierarchical groups, forming multilevel populations (genes in bacterial cells, chromosomes in eukaryotes, organisms in populations, populations in ecosystems, etc.). When the replication of particles (i.e. lower-level replicators) is not fully synchronized with the replication of the collective they belong to, or, in other words, when selective forces at different levels conflict, multilevel selection theory provides an effective description of the system [10,19]. A key ingredient of models of multilevel selection is the partitioning of fitness of particles to within-collective and between-collective components, once the collectives are defined. In particular, these models address the question of how cooperation between parts are selected for and maintained, against the 'selfish' within-collective replication of particles [20,21]. If selection on the collective level becomes so strong that individual replicators forfeit their autonomy, a transition in individuality takes place, forming a new, higher-level unit of evolution. Evolutionary transitions in individuality mark significant steps in life history on Earth, like the joining of genes into chromosomes, prokaryotes into the eukaryotic cell or individual cells into a multicellular organism [3,6]. Importantly, the identity of a new organism (a new level of individuality) consists of

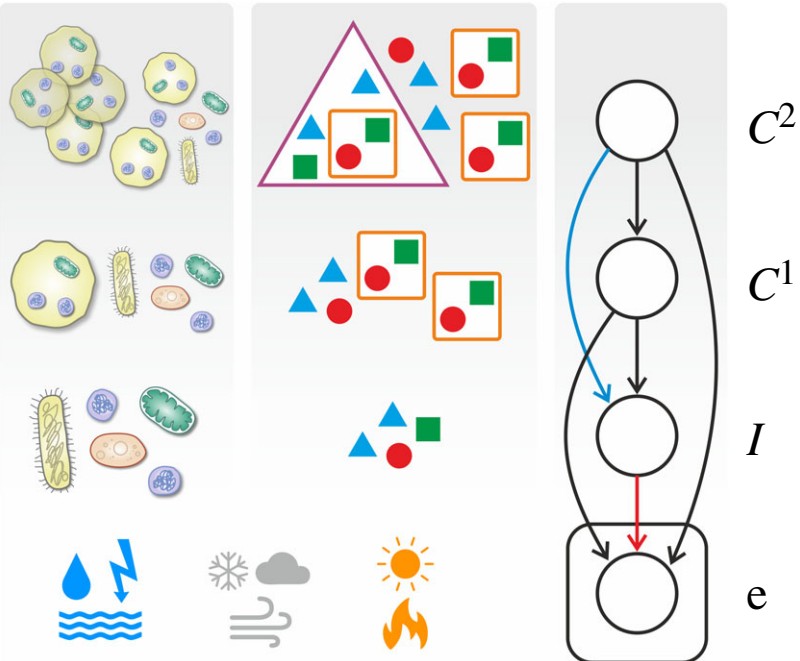

**Figure 1.** Evolution of multilevel population as inference in Bayesian belief network. The stochastic environment e governs the evolutionary dynamics of multilevel population composition $f\left(I_i \text{ in } C_j^1 \text{ in } C_k^2\right)$. This is, in turn, equivalent to successive Bayesian inference of hidden variables $I$, $C^1$ and $C^2$ based on the observation of current the environmental parameters e. Since these environmental parameters are sampled and observed multiple times (i.e. at every time step $t = 1, 2, 3, \ldots$), the corresponding node of the belief network is conventionally placed on a plate. Also note that the deletion of links between nodes of the belief network is corresponding to conditional independence relations between variables in the Bayesian setting and to specific structural properties of selection and population composition in the evolutionary setting; see text for details.

(i) inherited properties delivered by the replicators that form the group and (ii) emergent properties evolved newly within the group.

## 1.2. The equivalence of Bayesian update and replicator dynamics

In the following, we provide a brief introduction to the elementary building blocks of our arguments: Bayesian update and replicator dynamics. Bayesian update [22] fits a probability distribution $P(I)$ of hypotheses $I = I_1, \ldots, I_m$ to the data e. It does so by integrating prior knowledge about the probability $P(I_i)$ of hypothesis $I_i$ with the likelihood that the actual data $e = e(t)$ is being generated by hypothesis $I_i$, given by $P(e(t)|I_i)$. Mathematically, the fitted distribution $P(I_i|e(t))$, called the *posterior*, is simply proportional to both the *prior* $P(I_i)$ and the *likelihood* $P(e(t)|I_i)$:

$$P(I_i|e(t)) = \frac{P(e(t)|I_i)P(I_i)}{\sum_i P(e(t)|I_i)P(I_i)}. \tag{1.1}$$

On the other hand, the discrete replicator equation [23] that accounts for the change in relative abundance $f(I_i)$ of types of replicating individuals $I_i$ in the population driven by their fitness values $w(I_i)$, reads as

$$f(I_i; t+1) = \frac{w(I_i; t)f(I_i; t)}{\sum_i w(I_i; t)f(I_i; t)}. \tag{1.2}$$

As first noted by Harper [11] and Shalizi [12], equations (1.1) and (1.2) are equivalent, with the following identified quantities. The relative abundance $f(I_i; t)$ of type $I_i$ at time $t$ corresponds to the prior probability $P(I_i)$; the relative abundance $f(I_i; t+1)$ at time $t+1$ is corresponding to the posterior probability $P(I_i|e(t))$; the fitness $w(I_i; t)$ of type $I_i$ at time $t$ is corresponding to the likelihood $P(e(t)|I_i)$; and the *average fitness* $\sum_i w(I_i; t)f(I_i; t)$ is corresponding to the normalizing factor $\sum_i P(e(t)|I_i)P(I_i)$ called the *model evidence*.

Building on this observation, a natural question to ask is if this mathematical equivalence is only an apparent similarity due to the simplicity of both models, or it is a consequence of a deeper structural analogy between evolutionary and learning dynamics. We propose two conceptually new avenues

along which this equivalence can be generalized. First, we identify concepts of hierarchical evolutionary processes with concepts of (i) multivariate probability theory, (ii) Bayesian inference in hierarchical models and (iii) conditional independence relations between variables in such models. Building on this theoretical bridge, we then investigate the dynamics of learning the structure (as opposed to parameter fitting in a fixed model) of hierarchical Bayesian models and the Darwinian evolution of multilevel populations, concluding that following adaptive evolutionary paths on the landscape of hierarchical populations naturally maps to optimizing the structure of hierarchical Bayesian models via Bayesian model comparison.

## 2. Results

To generalize the algebraic equivalence between discrete-time replicator dynamics (equation (1.2)) and Bayesian update (equation (1.1)) to multilevel selection scenarios, multivariate distributions have to be involved. In general, a multivariate distribution $P(x_1, \ldots, x_k)$ over $k$ variables, each taking $m$ possible values, can be encoded by $m^k - 1$ independent parameters, which is exponential in the number of variables. Apart from practical considerations such as the possible infeasibility of computing marginal and conditional distributions, sampling and storing such general distributions, a crucial theoretical limitation is that fitting data by a model with such a sizable parameter space would result in overfitting, unless the training dataset is itself comparably large [24].

A way to overcome such obstacles is to explicitly abandon indirect dependencies between variables by using structured probabilistic models, such as belief networks (also called Bayesian networks or directed graphical models) [25,26]. Indeed, belief networks simplify joint distribution over multiple variables by specifying *conditional independence relations* corresponding to indirect (as opposed to direct) dependencies between variables.

In the following, we build up an algebraic isomorphism between discrete-time multilevel replicator dynamics and iterated Bayesian inference in belief networks on a step-by-step basis. The key identified quantities are summarized in table 1. This isomorphism is based on a mapping between the properties of multilevel populations and multivariate probability distributions, which we elaborate on in detail in the Methods section. Our first result extends the single-level replicator dynamics to the case of multilevel populations. Next, we explain how structural properties of multilevel selection of such populations map to the structure of Bayesian belief network, in particular regarding conditional independence relations. Finally, building on these steps, we discuss how evolutionary transitions in individuality can be interpreted as Bayesian structure learning.

### 2.1. Multilevel replicator dynamics as inference in Bayesian belief networks

Just like in the single-level case, the environmental parameters $e(t)$, $t = 1, 2, 3, \ldots$ are assumed to be sampled from an unknown generative process; the successive observation of them drives the successive update of population composition. As discussed earlier, however, multilevel population structures can be mapped to multivariate probability distributions, forming multiple *latent* variables $I, C^1, C^2, \ldots$ to be updated upon the observation of e.

Formally, just as prior probabilities over multiple hypotheses $P(I_i, C_j^1, C_k^2, \ldots; t)$ are updated to posterior probabilities $P(I_i, C_j^1, C_k^2, \ldots; t+1)$ based on the likelihood, $P(e(t) | I_i, C_j^1, C_k^2, \ldots; t)$, in the same way, multilevel population composition at time $t$, $f(I_i$ in $C_j^1$ in $C_k^2$ in $\ldots; t)$ is updated to the composition at $t+1$ based on fitnesses $w(I_i$ in $C_j^1$ in $C_k^2$ in $\ldots; t)$. The critical conceptual identification here is therefore of (i) the likelihood of the hypothesis parametrized by $(I_i, C_j^1, C_k^2, \ldots)$ and of (ii) the fitness of those individuals $I_i$ that belong to those collectives $C_j^1$ that belong to $C_k^2$, etc. The normalization factor that ensures that (i) the multivariate distribution is normalized (the model evidence $\sum_{i,j,k,\ldots} P(e(t) | I_i, C_j^1, C_k^2, \ldots; t) \times P(I_i, C_j^1, C_k^2, \ldots; t)$) or that (ii) abundances are always measured relative to the total abundance of individuals (the average fitness $\sum_{i,j,k,\ldots} w(I_i$ in $C_j^1$ in $C_k^2$ in $\ldots; t) \times f(I_i$ in $C_j^1$ in $C_k^2$ in $\ldots; t)$) is conceptually irrelevant here as they do not change the ratio of probabilities or abundances. Their equivalence will, however, play a critical role in relating evolution of individuality and structure learning of belief networks.

To demonstrate how simple calculations are performed in this framework and also to elucidate how fitnesses are determined, here we calculate the fitness of collective $C_j^1$, $w(C_j^1)$, which has been identified with $P(e|C_j^1)$. Using simple relations of probability theory, $P(e|C_j^1) = \sum_{I_i} P(e, I_i | C_j^1) = \sum_{I_i} P(e|I_i, C_j^1) P(I_i | C_j^1)$. Translating this back to the language of evolution tells us that the fitness of $C_j^1$ is

**Table 1.** Identified quantities of evolution and learning.

| multivariate probability theory | multilevel population |
|---|---|
| joint probabilities $P(l_i, C_j^1, C_k^2, \ldots)$ | relative abundances of individuals $f(l_i$ in $C_j^1$ in $C_k^2$ in $\ldots)$ |
| marginals, e.g. $P(C_j^1) = \sum_{i,k\ldots} P(l_i, C_j^1, C_k^2, \ldots)$ | relative abundances of units at a given level, e.g. of collectives at level $C^1$, $f(C_j^1) = \sum_{i,k\ldots} f(l_i$ in $C_j^1$ in $C_k^2$ in $\ldots) = f($any $l$ in $C_j^1)$ in any $C^2$ in $\ldots)$ |
| conditional probabilities, e.g. $P(l_i|C_j^1) = P(l_i, C_j^1)/P(C_j^1)$ or $P(C_j^1|l_i) = P(l_i, C_j^1)/P(l_i)$ | composition of collectives $f(l_i$ in $C_j^1)/f($any $l$ in $C_j^1)$ OR membership distribution of individuals $f(l_i$ in $C_j^1)/f(l_i$ in any $C^1)$ |
| **Bayesian inference in hierarchical models** | **multilevel replicator dynamics** |
| prior, $P(l_i, C_j^1, C_k^2 \ldots; t)$ | relative abundance $f(l_i$ in $C_j^1$ in $C_k^2$ in $\ldots; t)$ |
| likelihood, $P(e(t)|l_i, C_j^1, C_k^2 \ldots; t)$ | fitness $w(l_i$ in $C_j^1$ in $C_k^2$ in $\ldots; t)$ |
| posterior, $P(l_i, C_j^1, C_k^2, \ldots; t+1)$ | relative abundance $f(l_i$ in $C_j^1$ in $C_k^2$ in $\ldots; t+1)$ |
| model evidence, $\sum_{i,j,k\ldots} P(e(t)|l_i, C_j^1, C_k^2, \ldots; t) \times P(l_i, C_j^1, C_k^2, \ldots; t)$ | average fitness $\sum_{i,j,k\ldots} w(l_i$ in $C_j^1$ in $C_k^2$ in $\ldots; t) \times f(l_i$ in $C_j^1$ in $C_k^2$ in $\ldots; t)$ |
| **conditional independence relations** | **properties of multilevel selection** |
| conditional independence of the observed variable e and a latent variable, e.g. $l$, $P(e|l, C^1, C^2, \ldots) = P(e|C^1, C^2, \ldots)$ | units at a given level, e.g. individuals, 'freeze': their fitness is completely determined by the collective(s) they belong to: $w(l_i$ in $C_j^1$ in $C_k^2$ in $\ldots)$ is the same for all $i$ |
| conditional independence between two latent variables, e.g. $l$ and $C^2$, $P(l|C^1, C^2, \ldots) = P(l|C^1, \ldots)$ | the composition of units at level $C^1$ is independent of what units they belong to at level $C^2$. |
| **Bayesian structure learning** | **evolutionary transitions in individuality** |
| evidence of model $\mathcal{M}_a$, $E(\mathcal{M}_a) = P(e|\mathcal{M}_a) = \sum_{i,j,k\ldots} P(e|l_i, C_j^1, C_k^2, \ldots, \mathcal{M}_a) \times P(l_i, C_j^1, C_k^2, \ldots|\mathcal{M}_a)$ | average fitness given population composition $\mathcal{M}_a$, $\bar{w}(\mathcal{M}_a) = \sum_{i,j,k\ldots} w(l_i$ in $C_j^1$ in $C_k^2$ in $\ldots) \times f(l_i$ in $C_j^1$ in $C_k^2$ in $\ldots)$ |
| difference of evidence, $E(\mathcal{M}_b) - E(\mathcal{M}_a)$ | difference of average fitness of those units that are participating in the transition in individuality, causing the $\mathcal{M}_a \to \mathcal{M}_b$ change in population structure |

simply the average fitness of individuals being part of collective $C_j^1$, as anticipated earlier. Crucially, however, fitnesses of individuals depend on the identity of the collective they are part of. The fact that the fitness of a collective is computed as the average fitness of its individuals, therefore, does not constrain the way fitness of a collective emerges from the fitness and/or identity of its particles as being free. Modelling the evolutionary path toward an emerging identity of collectives is out of the scope of this paper; however, we point out that any such endeavour necessarily translates to coupling parameters at different levels of the Bayesian hierarchy.

## 2.2. Mapping structural properties of multilevel selection to the structure of Bayesian belief network

Structured probabilistic models are useful because they concisely summarize direct and indirect dependencies between multiple variables. Specifically, Bayesian belief networks depict multivariate distributions, such as $P(e,I,C^1,C^2)$, as a directed network, with the variables corresponding to the nodes and conditioning one variable on another corresponds to a directed link between the two. Since $P(e,I,C^1,C^2)$ can *always* be written as $P(e|I,C^1,C^2)P(I|C^1,C^2)P(C^1|C^2)P(C^2)$ in terms of conditional probabilities, the corresponding belief network is the one illustrated in figure 1. The route to simplify the structure of the distribution and correspondingly, the structure (i.e. connectivity) of the belief network is through *conditional independence relations*. Conditional independence relations, such as

$$P(e|I,C^1,C^2) = P(e|C^1,C^2), \qquad (2.1)$$

correspond to the deletion of connections; (2.1), for example, corresponds to the deletion of the connection between variables e and $I$, shown in red in figure 1, and it describes the conditional independence of the observed variable e and a latent variable, $I$. What does this independence relation mean in evolutionary terms? As it logically follows from the previous identifications, it specifies that the units at level $I$ are *frozen* in an evolutionary sense: their fitness is completely determined by the collective they belong to. There is a second, qualitatively different type of conditional independence relations: those between two latent variables, corresponding to two levels of the population. For example, $P(I|C^1,C^2) = P(I|C^1)$, corresponding to the deletion of the blue link in figure 1, is interpreted as the following: the composition of any collective at level $C^1$ is independent of what higher-level collective (at level $C^2$) it belongs to. Such simplifications in hierarchical population composition allow for the step-by-step modular combination of units to higher-level units, re-using existing sub-solutions over and over again.

## 2.3. Evolutionary transitions in individuality as Bayesian structure learning

It has been shown above that Bayesian inference in belief networks can be interpreted as Darwinian evolutionary dynamics of multilevel populations, driven by the 'observation' of the actual environment e($t$). What fits the environment is the hierarchical distribution of individuals (i.e. lowest level replicators) to collectives. However, the number of levels and the existing types within each level, along with the assumptions of hierarchical containment dependencies (i.e. conditional independence relations) has to be *a priori* specified. In this sense, fitting the environment by such a pre-defined structure via successive Bayesian updates has limited adaptation abilities. In particular, it is unable to adjust the complexity of the model to be in accordance with that of the environment, an inevitable property to avoid under- or overfitting.

To enlarge the space of possible models and therefore fit the environment better, one might allow the model structure to adapt as well (figure 2). More complex models, however, will *always* fit any data better, and accordingly, adapting the model structure naively might result in overfitting, i.e. the inability of the model to account for never-seen data, corresponding to possible future environments. Organisms with too complicated hierarchical containment structures (and other adaptive parameters that are not modelled explicitly here) would go extinct in any varying environment. To remedy this situation, one has to take into consideration not only how good the best parameter combination fits the data, but also how hard it is to find such a parameter combination. A systematic way of doing so is known as *Bayesian model comparison*, a well-known method in machine learning and Bayesian modelling. Mathematically, Bayesian model comparison simply ranks models (here, belief networks)

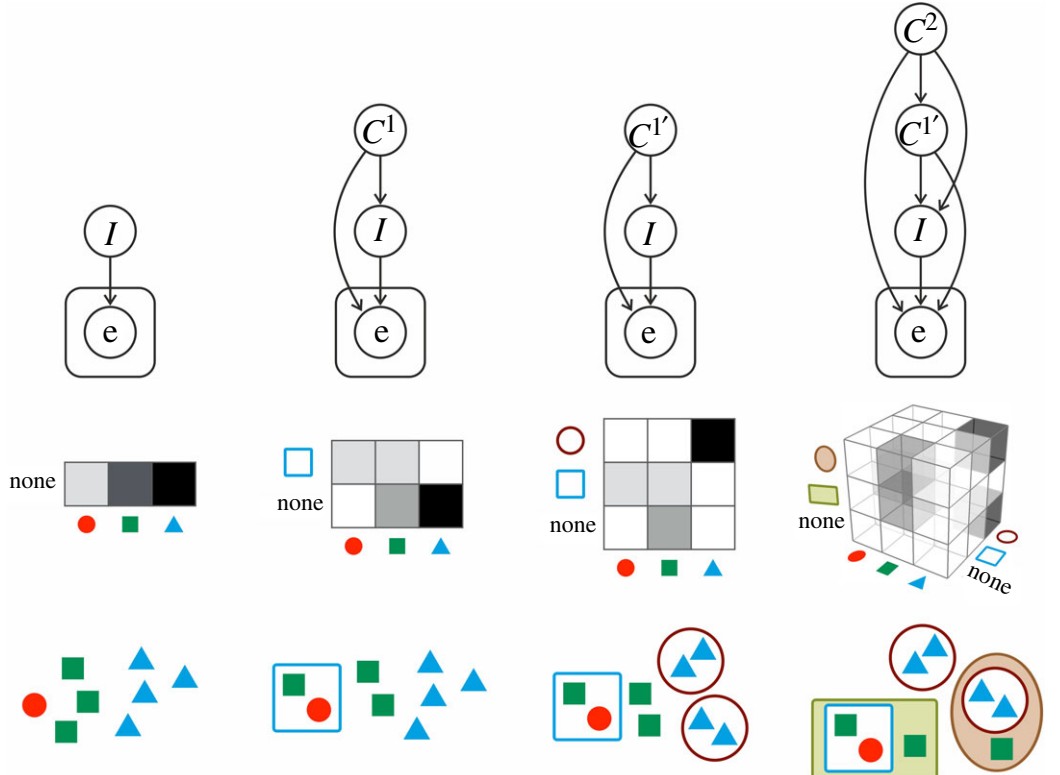

**Figure 2.** Evolutionary transitions as Bayesian structure learning. Initially, a single-level population $I$ fits the environment e via replicator dynamics, or equivalently, via successive Bayesian update. Then, a new collective (the square) emerges at a new level $C^1$, represented as a new node in the Bayesian belief network. Then, another new collective emerges at level $C^1$ (the circles), therefore, the variable $C^1$ is renamed to $C^{1'}$ as its possible values now include the circle as well. Finally, new collectives emerge at an even higher level (the rectangle and the ellipse at level $C^2$), and correspondingly, a new node is added to the network again. Note that the evolution of parameters (i.e. population composition in a fixed structure) is not illustrated here for simplicity.

according to their average ability to fit the data, referred to as the *evidence* $E(\mathcal{M})$ of model $\mathcal{M}$,

$$E(\mathcal{M}) = P(e|\mathcal{M}) = \sum_{i,j,k,\ldots} P(e|I_i, C_j^1, C_k^2, \ldots, \mathcal{M}) \times P(I_i, C_j^1, C_k^2, \ldots |\mathcal{M}). \tag{2.2}$$

The first term in the sum describes the likelihood of the current parameters (i.e. their ability to fit the data), whereas the second term weights these likelihoods according to the prior probabilities of the parameters.

How evolution, on the other hand, limits the number of to-be-fitted parameters in any organism to reinforce evolvability is an intriguing phenomenon. Here we show that in our minimal framework, selection naturally accounts for model complexity: model evidence corresponds to the average fitness $\bar{w}$ of individuals, determined by their hierarchical grouping to higher-level replicators. Indeed, interpreting equation (2.2) in evolutionary terms gives

$$\sum_{i,j,k,\ldots} w(I_i \text{ in } C_j^1 \text{ in } C_k^2 \text{ in } \ldots) \times f(I_i \text{ in } C_j^1 \text{ in } C_k^2 \text{ in } \ldots) = \bar{w}(\mathcal{M}), \tag{2.3}$$

in which the first term in the sum corresponds to fitnesses of individuals according to what collectives they belong to, and the second terms weights these fitnesses according to the abundance of such hierarchical arrangements. It implies that not only the evolution of the *composition* of multilevel population, but also the evolution of the *structure* of the multilevel population can be interpreted both in Darwinian and Bayesian terms: adaptive trajectories in the fitness landscape over population structures translate to adaptive trajectories of model evidence over belief networks. Note that the word structure here is borrowed from learning theory for consistency, and it does not refer to structured populations in population ecology.

Let us now turn specifically to the Bayesian interpretation of the evolution of individuality. Transitions in individuality, an evolutionary process in which lower-level units that were previously capable of independent replication form a higher-level evolutionary unit, correspond to specific type transitions in the Bayesian model structure: either a new node is added to the top of the network (in the case where there was no such population level at all earlier), or a new value is added to any of the existing variables (in the case where the new evolutionary unit is formed at an already existing level). In each case, most of the belief network, including its parameters, remains the same, except the part that is participating in the transition. This part, however, always involves only those values (corresponding to types) of those variables (corresponding to levels) that are participating in the transition. If average fitness of these types is larger by grouping them together, they undergo a transition in individuality. Although this is a general description of transitions disregarding many details, the correspondence with Bayesian model comparison is remarkable.

## 3. Discussion

Having defined our model framework mathematically, we now review its relation to multilevel selection and transition theory in more detail. Multilevel selection is conceptually characterized into two types, dubbed multilevel selection 1 (MLS1) and multilevel selection 2 (MLS2), both assuming that collectives form in a population of replicators, which themselves affect selection of lower-level units [6,10,19]. In the case of MLS1, only temporary collectives form that periodically disappear to revert to an unstructured population of lower-level units (transient compartmentation) [27,28]. MLS2, on the other hand, involves collectives that last and reproduce indefinitely, hence being bona fide evolutionary units [29], see also [30]). Only if collectives are evolutionary units can they inherit information stably (i.e. being informational replicators [31]), thus the step toward a major evolutionary transition is MLS2. Note that MLS1 can be understood as kin selection for most of the cases (cf. [29]), and might not even be a necessary prerequisite for MLS2 to evolve. In general, compartmentalization itself (transient or not) is not a sufficient property for a system to be a true evolutionary unit (cf. [32,33]).

Our framework allows for parametrization of collective fitnesses such that they only depend on the collective's composition, therefore corresponding to MLS2. The model is capable of handling MLS1 if, at each time step, individuals are randomly reassorted among higher-level collectives; incorporating this in the presented framework here is left for future work. Here we focus on the step from MLS2 toward a major transition: when collectives evolve to inherit information *above* their own composition. In our model, this corresponds to the case when a property of the collective appears, possibly assigning different identities to collectives having identical composition. Such an identity-providing piece of information is understood as an emergent property of the collective that does not depend on the composition of lower-level particles. If this is granted, higher-level units can evolve on their own, somewhat independent of their compositions. In biological context, such properties correspond either to novel epistatic interactions among genes or epigenetically inherited information that is not coded by genes.

## 4. Conclusion

In this paper, we introduced a mapping between concepts of hierarchical Bayesian models and concepts of Darwinian evolution, providing a learning theory-based interpretation of complexification of life through evolutionary transitions of individuality. The backbone of this interpretation is the fact that measuring the abundance and the composition of any type (organism, population, ecosystem, etc.) at any level can be naturally mapped to performing marginalization and computing conditional probabilities, respectively, of multivariate discrete probability distributions. Another key ingredient is that the stochastic environment determines the fitness of both individuals and collectives in a multilevel selection process. These two pillars are united by the already known algebraic equivalence between Bayesian update and discrete replicator dynamics. Accordingly, the learning theory narrative of multilevel selection is as follows: as the environment e is successively observed, the distribution over the latent variables $I, C^1, C^2, \ldots$, corresponding to the hierarchical population composition, is successively updated according to Bayes' rule.

Having identified this analogy, one might ask how the structure of the belief network (i.e. not just the parameters of a fixed network) itself evolves. In learning theory, different structures can be scored according to their model evidence, giving rise to Bayesian model comparison, which accounts not

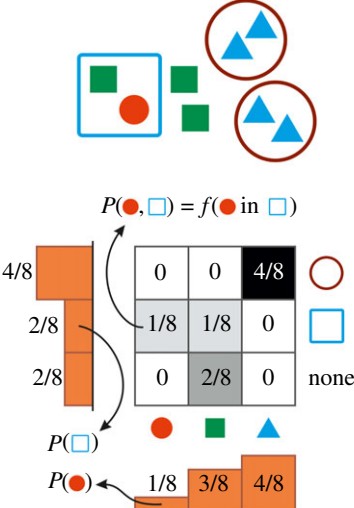

**Figure 3.** Two-level population encoded as a bivariate probability distribution. Joint probabilities represent the relative abundance of different individuals in different collectives. Conditional distributions depict the composition of collectives (rows) or the membership distribution of individuals (columns). Marginals, illustrated by the one-dimensional histograms, represent the abundance distribution of types at the individual level (horizontal) or at the level of collectives (vertical histogram).

only for how good a given solution is, but also for how unlikely it is to find such a good solution in the parameter space. Consequently, this procedure optimizes the trade-off between complexity and goodness of fit, hence dubbed as automatic Occam's razor. The evolution of belief network structure, in the context of Bayesian learning theory, is therefore driven by comparing model evidences of different structures. Interestingly, Bayesian model comparison fits neatly to our multilevel evolutionary dynamics interpretation: model evidence turns out to be equivalent to the average fitness of individuals, i.e. of the lowest level replicating units. This allows for a learning-theory-based view of evolutionary transitions in individuality: units aggregate to form a higher-level replicating unit if their average fitness increases by doing so; this is mathematically equivalent to performing Bayesian model comparison between the different belief network structures.

This procedure of simultaneous data acquisition, fitting and structure learning is far from unique to our proposed model framework; apart from its extensive use in machine learning algorithms, it is conjectured to govern classified-as-intelligent systems such as the conceptual development in children and also our collective understanding of the world in terms of scientific concepts, both relying on the extraordinary generalization abilities from sparse and noisy data [34,35]. We argue, based on the mathematical equivalence presented in this paper, that in order to devise seemingly engineered complex organisms, evolution on Earth or anywhere, used comparable hierarchical learning mechanisms as we humans do to make sense of the world around us.

# 5. Methods

Here we provide a mapping between properties of multilevel populations and multivariate probability theory. A multilevel population is regarded as a hierarchical containment structure of types: individual types $I_i$ might be part of collectives $C_j^1$ which themselves might be part of higher-level collectives $C_k^2$, and so on, as illustrated in figure 1. Note that collectives at any level might possess heritable information (henceforth referred to as their identity); collectives of the same (hierarchical) composition might very well have different identities. This makes this framework flexible enough to incorporate qualitatively different stages of evolutionary interdependence between organisms, leading eventually to a transition in individuality: (i) selection in which individuals enjoy the synergistic effect of belonging to a collective, but the collectives themselves do not possess any heritable information; (ii) selection in which collectives possess their own heritable information but also the individuals in them might replicate at different rates; and (iii) selection in which individuals have already lost their ability to replicate independently, therefore, their fitness is totally determined by the collective they belong to. As Michod & Nedelcu write [36, p. 61], 'group fitness is, initially, taken to be the average of the lower-level individual fitnesses; but as the evolutionary transition proceeds, group fitness

becomes decoupled from the fitness of its lower-level components'. This, as we shall see, is exactly what our model accounts for mathematically, incorporating also the effect of stochastically varying environment.

A key assumption that enables the machinery of multivariate probability theory to work is that abundance of collectives is measured in terms of abundance of individuals they contain. Indeed, by identifying the abundance of individuals of type $I_i$, $f(I_i$ in $C_j^1$ in $C_k^2$ in $\ldots)$, that are part of collectives of type $C_j^1$ that are themselves part of collectives of type $C_k^2$, etc., with the joint probabilities $P(I_i, C_j^1, C_k^2, \ldots)$, two important additional identifications follow:

— *marginal distributions*, such as $P(C_j^1) = \sum_{i,k,\ldots} P(I_i, C_j^1, C_k^2, \ldots)$ translate to the *abundance distribution of types at the corresponding level* (here, $C^1$), $f(C_j^1) = \sum_{i,k,\ldots} f(I_i$ in $C_j^1$ in $C_k^2$ in $\ldots) = f(\text{any } I$ in $C_j^1$ in any $C^2$ in $\ldots)$;

— *conditional distributions*, e.g. $P(I_i|C_j^1) = P(I_i, C_j^1)/P(C_j^1)$ or $P(C_j^1|I_i) = P(I_i, C_j^1)/P(I_i)$ translate either to *composition of collectives* $f(I_i$ in $C_j^1)/f(\text{any } I$ in $C_j^1)$ or *membership distribution of individuals* (or lower-level collectives), $f(I_i$ in $C_j^1)/f(I_i$ in any $C^1)$.

These computations are illustrated by a toy example in figure 3. The rest of our methodology forms an integral part of our results hence it is explained under Results.

Data accessibility. This article has no additional data.

Authors' contributions. D.C. designed the mathematical formalism and performed analysis. I.Z. contributed to the conceptualization of the theoretical framework. E.S. conceived, funded and supervised the research. D.C. wrote the initial draft, I.Z. and E.S. contributed and reviewed the manuscript. All authors read and approved the final manuscript.

Competing interests. We declare we have no competing interests.

Funding. The authors acknowledge financial support from the National Research, Development, and Innovation Office under NKFI-K119347 (E.S.) NKFI-K124438 (I.Z.), 'Theory and solutions in the light of evolution' GINOP-2.3.2-15-2016-00057 (D.C., I.Z. and E.S.); the Volkswagen Stiftung initiative 'Leben? – Ein neuer Blick der Naturwissenschaften auf die grundlegenden Prinzipien des Lebens' under project 'A unified model of recombination in life' (E.S.); and the Templeton World Charity Foundation under grant number TWCF0268 (D.C. and E.S.). These funding bodies did not have a role in the design of this analysis and in the interpretation of our results, and in writing the manuscript.

Acknowledgements. The authors thank Szabolcs Számadó, Ádám Radványi, András Szilágyi, András Hubai and the three reviewers for their insightful comments on the manuscript.

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
