## [Reviewer comments · Royal Society Open Science]

Review History

RSOS-190202.R0 (Original submission)

Review form: Reviewer 1 (Steven A. Frank)

Is the manuscript scientifically sound in its present form?

Yes

Are the interpretations and conclusions justified by the results?

Yes

Is the language acceptable?

Yes

Is it clear how to access all supporting data?

Not Applicable

Do you have any ethical concerns with this paper?

No

Have you any concerns about statistical analyses in this paper?

No

Recommendation?

Accept as is

Comments to the Author(s)

It has often been suggested that natural selection is similar to an algorithm that learns about the environment. Further analogies have related natural selection to information theory and to Bayesian inference. Although those analogies seem broadly sensible, the highly developed formal theory of natural selection is very different from formal theories of information and Bayesian inference.

Is the analogy just a vague similarity? Or can one usefully think of natural selection as part of a broader formal theory of learning and inference? If so, what exactly is the relation? Does the formal relation provide insight into how natural selection has shaped the evolutionary history of life?

This article takes a significant step toward relating natural selection and theories of learning and inference. The authors focus on evolutionary transitions in biology, in which hierarchical complexity develops over time. They show a possible formal relation between the evolution of hierarchical complexity by natural selection and hierarchical Bayesian algorithms that recursively combine subsolutions to achieve solutions to higher-level problems.

Perhaps the greatest value of this work is to show the potential for further studies. Computational disciplines are making great progress in the analysis of learning for different kinds of challenges. Evolutionary history in biology provides a rich empirical source of how an algorithmic process has actually learned about complex environmental challenges. The joint study of learning theory and evolution is likely to provide significant insight to both subjects.

I do not have specific comments about the details of the manuscript. The approach is reasonable. But we will have to wait for further studies that explore alternative ways to relate learning theories and evolutionary theories before we can evaluate the best approaches, potential conceptual insights, and future directions.

Steven Frank

Review form: Reviewer 2**Is the manuscript scientifically sound in its present form?**

Yes

Are the interpretations and conclusions justified by the results?

Yes

Is the language acceptable?

Yes

Is it clear how to access all supporting data?

Yes

Do you have any ethical concerns with this paper?

No

Have you any concerns about statistical analyses in this paper?

No

Recommendation?

Accept as is

Comments to the Author(s)

It is well-known that there is a mathematical isomorphism between replicator dynamics, often used to formalize the process of evolution by natural selection, and Bayesian updating. (Whether this is a curiosity, or indicative of a deep conceptual connection, is of course another matter.) The authors aim to generalize this well-known point, by pointing to an (apparent) connection between multi-level selection and Bayesian inference in belief networks.

The paper does a good job of pointing to the isomorphism between these two bodies of theory, which is certainly an interesting point. (Whether it shows anything deep is another matter.) The application to evolutionary transitions, while speculative, is interesting.

The only downside of the paper is that not many readers are likely to be well-versed in both the theory of evolutionary transitions in biology, and the theory of Bayesian model comparison in statistics. So the readership for the paper is perhaps somewhat limited. That said, the conceptual and formal connections that the paper makes are interesting, and worth pointing out.

Review form: Reviewer 3 (Günter Wagner)

Is the manuscript scientifically sound in its present form?

Yes

Are the interpretations and conclusions justified by the results?

Yes

Is the language acceptable?

Yes

Is it clear how to access all supporting data?

Not Applicable

Do you have any ethical concerns with this paper?

No

Have you any concerns about statistical analyses in this paper?

No

Recommendation?

Accept with minor revision (please list in comments)

Comments to the Author(s)

This is an extraordinarily important paper, that has the potential to be correspondingly obscure to the unprepared reader. But its general message is quite straight forward and conceptually important: the dynamics of how natural selection is structuring natural populations is mathematically and, so the claim of the authors, also conceptually equivalent to Bayesian network learning. Let me start with a few words of background:

The idea that there are deep relationships between evolution by natural selection and learning by organisms is relatively old. This also extends to the core insight of this paper, namely that the structure of complex organisms can be understood in analogy of concept learning, such that a body plan of an organism can be understood as a particular “theory” of how the, for the species, relevant parts of the world is causally structured. This idea is, for instance, the core of Rupert Riedl’s “systems theory of evolution” (1975, English in 1978). Having said that, to my knowledge this analogy has never been developed as clearly and with as firm a mathematical foundation as in this paper.

Making this connection between natural selection theory and structured Bayesian learning theory, both formally as well as conceptually [which is not the same], is an important conceptual step that will exercise researchers for decades to come, in order to work out all the consequences of this idea.

Not surprisingly, a paper that makes such an important step has a tendency of being obscure at places. Here are a few hints of how to improve the accessibility of the paper.

Overall the paper would benefit if multi-level selection theory would be introduced without reference to the analogy to Bayesian models, and the same for Bayesian models, and only then discussing the relationship between the two. As it stands, the narrative goes forth and back between the two and the reader not equally familiar with both has a hard time to follow the argument.

P2, Line 10: in what sense are the authors speaking of “pre-adapted units”. I think I know what they mean, but the usual meaning of this term is somewhat oblique to what it means here.

P2, Line 18: “extreme form of cooperation” not clear what “extreme” means?

P3, Line 14: the discrete selection equation goes back to Fisher and Wright, not Martin Nowak, as Martin himself will assert.

P4, Line 27: the symbols “ C_i^n ” are introduced without explanation.

P4, line 48: here the authors seem to say that the mean fitness of the collective is the mean fitness of the individuals it contains, but that seems to suggest that the collective does not have emergent properties, which is contradicted later in the paper. Something is not clear in that statement. It could mean that the mean fitness of the collective is in fact the mean fitness of the individuals AS THEY EXPERIENCE SELECTION BEING IN THAT COLLECTIVE. Please clarify.

The last paragraph on page 6 is obscure, since it seems that the predefinition of the base functions for the environmental state distribution has to somehow be related to the nature of the collectives they are assigned to... may be this discussion needs to be expanded or deleted.

P7, line 34: jumping to the conclusion that these effects have to be epigenetic seems premature, given that one can think of scenarios where the effect is due to epistatic interactions rather than epigenetic factors.

Günter P. Wagner
Yale University

Decision letter (RSOS-190202.R0)

08-May-2019

Dear Mr Czégel,

The editors assigned to your paper ("Multilevel selection as Bayesian inference, major transitions in individuality as structure learning") have now received comments from reviewers. We would like you to revise your paper in accordance with the referee and Associate Editor suggestions which can be found below (not including confidential reports to the Editor). Please note this decision does not guarantee eventual acceptance.

Please submit a copy of your revised paper before 31-May-2019. Please note that the revision deadline will expire at 00.00am on this date. If we do not hear from you within this time then it will be assumed that the paper has been withdrawn. In exceptional circumstances, extensions may be possible if agreed with the Editorial Office in advance. We do not allow multiple rounds of revision so we urge you to make every effort to fully address all of the comments at this stage. If deemed necessary by the Editors, your manuscript will be sent back to one or more of the original reviewers for assessment. If the original reviewers are not available, we may invite new reviewers.

- Data accessibility

<http://datadryad.org/submit?journalID=RSOS&manu=RSOS-190202>

- Competing interests

- Authors' contributions

- Acknowledgements

- Funding statement

Kind regards,

Andrew Dunn

on behalf of Prof Kevin Padian (Subject Editor)
openscience@royalsociety.org

Associate Editor's comments:

Please respond to the comments of the reviewers in your revision and provide a thorough explanation (and any rebuttals) in the point-by-point response you submit.

Comments to Author:

Reviewers' Comments to Author:

Reviewer: 1

Comments to the Author(s)

It has often been suggested that natural selection is similar to an algorithm that learns about the environment. Further analogies have related natural selection to information theory and to Bayesian inference. Although those analogies seem broadly sensible, the highly developed formal theory of natural selection is very different from formal theories of information and Bayesian inference.

Is the analogy just a vague similarity? Or can one usefully think of natural selection as part of a broader formal theory of learning and inference? If so, what exactly is the relation? Does the formal relation provide insight into how natural selection has shaped the evolutionary history of life?

This article takes a significant step toward relating natural selection and theories of learning and inference. The authors focus on evolutionary transitions in biology, in which hierarchical complexity develops over time. They show a possible formal relation between the evolution of hierarchical complexity by natural selection and hierarchical Bayesian algorithms that recursively combine subsolutions to achieve solutions to higher-level problems.

Perhaps the greatest value of this work is to show the potential for further studies. Computational disciplines are making great progress in the analysis of learning for different kinds of challenges. Evolutionary history in biology provides a rich empirical source of how an algorithmic process has actually learned about complex environmental challenges. The joint study of learning theory and evolution is likely to provide significant insight to both subjects.

I do not have specific comments about the details of the manuscript. The approach is reasonable. But we will have to wait for further studies that explore alternative ways to relate learning theories and evolutionary theories before we can evaluate the best approaches, potential conceptual insights, and future directions.

Steven Frank

Reviewer: 2

Comments to the Author(s)

It is well-known that there is a mathematical isomorphism between replicator dynamics, often used to formalize the process of evolution by natural selection, and Bayesian updating. (Whether this is a curiosity, or indicative of a deep conceptual connection, is of course another matter.) The authors aim to generalize this well-known point, by pointing to an (apparent) connection between multi-level selection and Bayesian inference in belief networks.

The paper does a good job of pointing to the isomorphism between these two bodies of theory, which is certainly an interesting point. (Whether it shows anything deep is another matter.) The application to evolutionary transitions, while speculative, is interesting.

The only downside of the paper is that not many readers are likely to be well-versed in both the theory of evolutionary transitions in biology, and the theory of Bayesian model comparison in statistics. So the readership for the paper is perhaps somewhat limited.

That said, the conceptual and formal connections that the paper makes are interesting, and worth pointing out.

Reviewer: 3

Comments to the Author(s)

This is an extraordinarily important paper, that has the potential to be correspondingly obscure to the unprepared reader. But its general message is quite straight forward and conceptually important: the dynamics of how natural selection is structuring natural populations is mathematically and, so the claim of the authors, also conceptually equivalent to Bayesian network learning. Let me start with a few words of background:

The idea that there are deep relationships between evolution by natural selection and learning by organisms is relatively old. This also extends to the core insight of this paper, namely that the structure of complex organisms can be understood in analogy of concept learning, such that a body plan of an organism can be understood as a particular “theory” of how the, for the species, relevant parts of the world is causally structured. This idea is, for instance, the core of Rupert Riedl’s “systems theory of evolution” (1975, English in 1978). Having said that, to my knowledge this analogy has never been developed as clearly and with as firm a mathematical foundation as in this paper.

Making this connection between natural selection theory and structured Bayesian learning theory, both formally as well as conceptually [which is not the same], is an important conceptual step that will exercise researchers for decades to come, in order to work out all the consequences of this idea.

Not surprisingly, a paper that makes such an important step has a tendency of being obscure at places. Here are a few hints of how to improve the accessibility of the paper.

Overall the paper would benefit if multi-level selection theory would be introduced without reference to the analogy to Bayesian models, and the same for Bayesian models, and only then discussing the relationship between the two. As it stands, the narrative goes forth and back between the two and the reader not equally familiar with both has a hard time to follow the argument.

P2, Line 10: in what sense are the authors speaking of “pre-adapted units”. I think I know what they mean, but the usual meaning of this term is somewhat oblique to what it means here.

P2, Line 18: “extreme form of cooperation” not clear what “extreme” means?

P3, Line 14: the discrete selection equation goes back to Fisher and Wright, not Martin Nowak, as Martin himself will assert.

P4, Line 27: the symbols “ C_i^n ” are introduced without explanation.

P4, line 48: here the authors seem to say that the mean fitness of the collective is the mean fitness of the individuals it contains, but that seems to suggest that the collective does not have emergent properties, which is contradicted later in the paper. Something is not clear in that statement. It could mean that the mean fitness of the collective is in fact the mean fitness of the individuals AS THEY EXPERIENCE SELECTION BEING IN THAT COLLECTIVE. Please clarify.

The last paragraph on page 6 is obscure, since it seems that the predefinition of the base functions for the environmental state distribution has to somehow be related to the nature of the collectives they are assigned to... may be this discussion needs to be expanded or deleted.

P7, line 34: jumping to the conclusion that these effects have to be epigenetic seems premature, given that one can think of scenarios where the effect is due to epistatic interactions rather than epigenetic factors.

Günter P. Wagner
Yale University

Author's Response to Decision Letter for (RSOS-190202.R0)

See Appendix A.

RSOS-190202.R1 (Revision)

Review form: Reviewer 3 (Günter Wagner)

Is the manuscript scientifically sound in its present form?

Yes

Are the interpretations and conclusions justified by the results?

Yes

Is the language acceptable?

Yes

Is it clear how to access all supporting data?

Not Applicable

Do you have any ethical concerns with this paper?

No

Have you any concerns about statistical analyses in this paper?

No

Recommendation?

Accept with minor revision (please list in comments)

Comments to the Author(s)

This revision of the original submission is much better to read and much clearer. This was achieved by introducing different concepts one at a time and then discuss their relationship, as well as eliminating topics that can not be explained thoroughly in this short paper. I think this version is ready for prime time.

There is only one small thing that I miss in this version and that is the following: the authors now have a section dedicated to multilevel selection, but this section is entirely verbal. I do not think that is what was asked for. Since the argument in this paper is mainly based on structural similarities among mathematical models it would be nice to see the generic structure of a multi-level selection model explained in this section.

Otherwise I maintain what I said in my previous review: this is an important paper that will stimulate a lot of follow up work.

Günter P. Wagner, Yale University

Decision letter (RSOS-190202.R1)

22-Jul-2019

Dear Mr Czégel:

On behalf of the Editors, I am pleased to inform you that your Manuscript RSOS-190202.R1 entitled "Multilevel selection as Bayesian inference, major transitions in individuality as structure learning" has been accepted for publication in Royal Society Open Science subject to minor revision in accordance with the referee suggestions. Please find the referees' comments at the end of this email.

The reviewers and Subject Editor have recommended publication, but also suggest some minor revisions to your manuscript. Therefore, I invite you to respond to the comments and revise your manuscript.

- Ethics statement

- Data accessibility

If you wish to submit your supporting data or code to Dryad (<http://datadryad.org/>), or modify your current submission to dryad, please use the following link:
<http://datadryad.org/submit?journalID=RSOS&manu=RSOS-190202.R1>

- **Competing interests**

- **Authors' contributions**

- **Acknowledgements**

- **Funding statement**

Because the schedule for publication is very tight, it is a condition of publication that you submit the revised version of your manuscript before 31-Jul-2019. Please note that the revision deadline will expire at 00.00am on this date. If you do not think you will be able to meet this date please let me know immediately.

When submitting your revised manuscript, you will be able to respond to the comments made by the referees and upload a file "Response to Referees" in "Section 6 - File Upload". You can use this to document any changes you make to the original manuscript. In order to expedite the

processing of the revised manuscript, please be as specific as possible in your response to the referees.

on behalf of Prof Kevin Padian (Subject Editor)
openscience@royalsociety.org

Reviewer comments to Author:
Reviewer: 3

Comments to the Author(s)

This revision of the original submission is much better to read and much clearer. This was achieved by introducing different concepts one at a time and then discuss their relationship, as well as eliminating topics that can not be explained thoroughly in this short paper. I think this version is ready for prime time.

There is only one small thing that I miss in this version and that is the following: the authors now have a section dedicated to multilevel selection, but this section is entirely verbal. I do not think that is what was asked for. Since the argument in this paper is mainly based on structural

similarities among mathematical models it would be nice to see the generic structure of a multi-level selection model explained in this section.

Otherwise I maintain what I said in my previous review: this is an important paper that will stimulate a lot of follow up work.

Günter P. Wagner, Yale University

Author's Response to Decision Letter for (RSOS-190202.R1)

See Appendix B.

Decision letter (RSOS-190202.R2)

25-Jul-2019

Dear Mr Czégel,

I am pleased to inform you that your manuscript entitled "Multilevel selection as Bayesian inference, major transitions in individuality as structure learning" is now accepted for publication in Royal Society Open Science.

Kind regards,

on behalf of Kevin Padian (Subject Editor)
openscience@royalsociety.org

Appendix A

Dear Andrew Dunn,

Thank you for considering our manuscript for publication in Royal Society Open Science. Please find point-by-point responses to the reviewers' comments below.

Reviewer: 1

Comments to the Author(s)

It has often been suggested that natural selection is similar to an algorithm that learns about the environment. Further analogies have related natural selection to information theory and to Bayesian inference. Although those analogies seem broadly sensible, the highly developed formal theory of natural selection is very different from formal theories of information and Bayesian inference.

Is the analogy just a vague similarity? Or can one usefully think of natural selection as part of a broader formal theory of learning and inference? If so, what exactly is the relation? Does the formal relation provide insight into how natural selection has shaped the evolutionary history of life?

This article takes a significant step toward relating natural selection and theories of learning and inference. The authors focus on evolutionary transitions in biology, in which hierarchical complexity develops over time. They show a possible formal relation between the evolution of hierarchical complexity by natural selection and hierarchical Bayesian algorithms that recursively combine subsolutions to achieve solutions to higher-level problems.

Perhaps the greatest value of this work is to show the potential for further studies. Computational disciplines are making great progress in the analysis of learning for different kinds of challenges. Evolutionary history in biology provides a rich empirical source of how an algorithmic process has actually learned about complex environmental challenges. The joint study of learning theory and evolution is likely to provide significant insight to both subjects.

I do not have specific comments about the details of the manuscript. The approach is reasonable. But we will have to wait for further studies that explore alternative ways to relate learning theories and evolutionary theories before we can evaluate the best approaches, potential conceptual insights, and future directions.

Steven Frank

We thank Professor Frank for the generous and insightful review. We agree with the statement that the greatest value of this manuscript is to point at potential future directions, and that many details/complementary ideas have to be established in order to achieve fruitful idea transfer between the fields of evolutionary and learning theory.

Reviewer: 2

Comments to the Author(s)

It is well-known that there is a mathematical isomorphism between replicator dynamics, often used to formalize the process of evolution by natural selection, and Bayesian updating. (Whether this is a curiosity, or indicative of a deep conceptual connection, is of course another matter.)

The authors aim to generalize this well-known point, by pointing to an (apparent) connection between multi-level selection and Bayesian inference in belief networks.

The paper does a good job of pointing to the isomorphism between these two bodies of theory, which is certainly an interesting point. (Whether it shows anything deep is another matter.)

The application to evolutionary transitions, while speculative, is interesting.

The only downside of the paper is that not many readers are likely to be well-versed in both the theory of evolutionary transitions in biology, and the theory of Bayesian model comparison in statistics. So the readership for the paper is perhaps somewhat limited.

That said, the conceptual and formal connections that the paper makes are interesting, and worth pointing out.

We thank Reviewer 2 for the positive review. We agree that i) finding a deeper philosophical or mathematical principles behind these mathematical equivalences is an open and important question and also that ii) the application to evolutionary transitions is speculative and only provides a first approximation to any actual biological process.

The issue of potentially limited readership due to a relatively high level of presupposed knowledge in both Bayesian statistics and multilevel selection theory/transitions theory has also been pointed out by Reviewer 3 (Prof. Wagner), we reply to the corresponding comments jointly below.

Reviewer: 3

Comments to the Author(s)

This is an extraordinarily important paper, that has the potential to be correspondingly obscure to the unprepared reader. But its general message is quite straight forward and conceptually important: the dynamics of how natural selection is structuring natural populations is mathematically and, so the claim of the authors, also conceptually equivalent to Bayesian network learning. Let me start with a few words of background:

The idea that there are deep relationships between evolution by natural selection and learning by organisms is relatively old. This also extends to the core insight of this paper, namely that the structure of complex organisms can be understood in analogy of concept learning, such that a body plan of an organism can be understood as a particular “theory” of how the, for the species, relevant parts of the world is causally structured. This idea is, for instance, the core of Rupert Riedl’s “systems theory of evolution” (1975, English in 1978). Having said that, to my knowledge this analogy has never been developed as clearly and with as firm a mathematical foundation as in this paper.

Making this connection between natural selection theory and structured Bayesian learning theory, both formally as well as conceptually [which is not the same], is an important conceptual step that will exercise researchers for decades to come, in order to work out all the consequences of this idea.

Not surprisingly, a paper that makes such an important step has a tendency of being obscure at places. Here are a few hints of how to improve the accessibility of the paper.

Overall the paper would benefit if multi-level selection theory would be introduced without reference to the analogy to Bayesian models, and the same for Bayesian models, and only then discussing the relationship between the two. As it stands, the narrative goes forth and back between the two and the reader not equally familiar with both has a hard time to follow the argument.

P2, Line 10: in what sense are the authors speaking of “pre-adapted units”. I think I know what they mean, but the usual meaning of this term is somewhat oblique to what it means here.

P2, Line 18: “extreme form of cooperation” not clear what “extreme” means?

P3, Line 14: the discrete selection equation goes back to Fisher and Wright, not Martin Nowak, as Martin himself will assert.

P4, Line 27: the symbols “ C_i^n ” are introduced without explanation.

P4, line 48: here the authors seem to say that the mean fitness of the collective is the mean fitness of the individuals it contains, but that seems to suggest that the collective does not have emergent properties, which is contradicted later in the paper. Something is not clear in that statement. It could mean that the mean fitness of the collective is in fact the mean fitness of the individuals AS THEY EXPERIENCE SELECTION BEING IN THAT COLLECTIVE. Please clarify.

The last paragraph on page 6 is obscure, since it seems that the predefinition of the base functions for the

environmental state distribution has to somehow be related to the nature of the collectives they are assigned to... may be this discussion needs to be expanded or deleted.

P7, line 34: jumping to the conclusion that these effects have to be epigenetic seems premature, given that one can think of scenarios where the effect is due to epistatic interactions rather than epigenetic factors.

Günter P. Wagner
Yale University

We thank Professor Wagner for providing a thorough review and acknowledge both the general recommendations regarding possible ways to improve accessibility and the specific comments on the manuscript.

In order to improve accessibility, we added a subsection to Introduction, titled “Darwinian evolution of multilevel populations”, where crucial points of multilevel selection and transitions in individuality are briefly discussed without reference to Bayesian models. We also moved existing text introducing Bayesian update and replicator dynamics to a new subsection titled “The equivalence of Bayesian update and replicator dynamics” to increase the clarity of the structure of the manuscript.

Although we agree that a separate introduction to Bayesian concepts would be helpful to the reader, we believe that given length considerations, the brief conceptual and mathematical introduction to each relevant Bayesian concept that is provided at the beginning of each subsection in Methods serves as a good starting point to the interested but unfamiliar reader.

We address the specific points below.

P2, Line 10: in what sense are the authors speaking of “pre-adapted units”. I think I know what they mean, but the usual meaning of this term is somewhat oblique to what it means here.

In order to avoid confusion and decrease redundancy, we deleted the second part of the sentence “via hierarchical aggregation of pre-adapted subunits”.

Therefore, the sentence

“Such *evolutionary transitions in individuality* opened the door to the vast increase of complexity via hierarchical aggregation of pre-adapted subunits.”

Has been changed to

“Such *evolutionary transitions in individuality* opened the door to the vast increase of complexity.”

P2, Line 18: “extreme form of cooperation” not clear what “extreme” means?

It is indeed misleading and unnecessary. We deleted “extreme form of”. We also changed the beginning of the last sentence of that paragraph from “At that point, ...” to “At any phase” since multilevel selection theory is the relevant at any phase of the transition process.

P3, Line 14: the discrete selection equation goes back to Fisher and Wright, not Martin Nowak, as Martin himself will assert.

We now cite (Wright, 1931), citation 24.

Line 27: the symbols “ C_i^n ” are introduced without explanation.

We now introduce the notation in the previous sentence by adding “ I, C^1, C^2, \dots ”
The sentence in this update form reads as:

“As discussed earlier, however, multilevel population structures can be mapped to multivariate probability distributions, forming multiple *latent* variables I, C^1, C^2, \dots to be updated upon the observation of e .”

P4, line 48: here the authors seem to say that the mean fitness of the collective is the mean fitness of the individuals it contains, but that seems to suggest that the collective does not have emergent properties, which is contradicted later in the paper. Something is not clear in that statement. It could mean that the mean fitness of the collective is in fact the mean fitness of the individuals AS THEY EXPERIENCE SELECTION BEING IN THAT COLLECTIVE. Please clarify.

This is a crucial yet highly non-trivial issue, thank you for pointing it out. We made the following changes to clarify this point.

1. We substituted “it contains” to “being part of collective C_j^{1*} ” in the referred sentence, which now reads as:

“Translating this back to the language of evolution tells us that the fitness of C_j^{1*} is simply the average fitness of individuals being part of collective C_j^{1*} , as anticipated earlier.”

2. We added the following explanatory sentences to the end of the referred paragraph:

“Crucially, however, fitnesses of individuals depend on the identity of the collective they are part of. The fact that the fitness of a collective is computed as the average fitness of its individuals therefore does not constrain the way fitness of a collective emerges from the fitness and/or identity of its particles as being free. Modeling the evolutionary path toward an emerging identity of collectives is out of the scope of this paper, however, we point out that any such endeavour necessarily translates to coupling parameters at different levels of the Bayesian hierarchy.”

We hope that these modifications make the argument more clear.

The last paragraph on page 6 is obscure, since it seems that the predefinition of the base functions for the environmental state distribution has to somehow be related to the nature of the collectives they are assigned to... may be this discussion needs to be expanded or deleted.

It is indeed obscure, and clarifying explanations would need many more points to be discussed, points that should be built on additional structures that are not introduced in this manuscript. We therefore deleted the paragraph, as it is tangential to the main line of reasoning here. (We discuss many of these points in a subsequent manuscript.)

P7, line 34: jumping to the conclusion that these effects have to be epigenetic seems premature, given that one can think of scenarios where the effect is due to epistatic interactions rather than epigenetic factors.

We agree. We now emphasize epigenetic effects as well. The updated sentence is

“In biological context, such properties correspond either to novel epistatic interactions among genes or epigenetically inherited information that is not coded by genes.”

Sincerely,
Dániel Czégel
on behalf of the authors

Appendix B

Dear Andrew Dunn,

Thank you very much for accepting our manuscript for publication in Royal Society Open Science.

Please find our responses to the referee's comments as well as our edits to the manuscript below.

Comments to the Author(s)

This revision of the original submission is much better to read and much clearer. This was achieved by introducing different concepts one at a time and then discuss their relationship, as well as eliminating topics that can not be explained thoroughly in this short paper. I think this version is ready for prime time.

There is only one small thing that I miss in this version and that is the following: the authors now have a section dedicated to multilevel selection, but this section is entirely verbal. I do not think that is what was asked for. Since the argument in this paper is mainly based on structural similarities among mathematical models it would be nice to see the generic structure of a multi-level selection model explained in this section.

Otherwise I maintain what I said in my previous review: this is an important paper that will stimulate a lot of follow up work.

Günter P. Wagner, Yale University

We thank professor Wagner for the positive and constructive comments.

We believe that introducing one specific model of multilevel selection might suggest to the reader that this paper is about a mathematical equivalence between Bayesian inference in hierarchical graphical models and that specific model of multilevel selection. Here, however, what we want is to raise the attention to is a rather general relation between these two concepts with many further limitations to be discussed in the future as well as many details and possible extensions to be worked out.

We therefore chose to refer to a general feature of mathematical models of multilevel selection in the introduction but did not introduce any specific formulations: in the section "Darwinian evolution of multilevel populations", we added the sentence

"A key ingredient of models of multilevel selection is the partitioning of fitness of particles to within-collective and between-collective components, once the collectives are defined. "

Furthermore, as a minor change in the same section, in order to align our introduction to multilevel selection with the mostly used nomenclature of multilevel selection theory, we refer to the individual replicators as "particles" instead of "parts". In order to also make it accessible to a general readership, we add "i.e., lower level replicators" in parentheses after the first mention of "particle".

We hope that these minor changes further improve the clarity of the paper.

Sincerely,
Dániel Czégel
on behalf of the authors